# Evaluation of Ascorbic Acid or Curcumin Formulated in a Solid Dispersion on *Salmonella* Enteritidis Infection and Intestinal Integrity in Broiler Chickens

**DOI:** 10.3390/pathogens8040229

**Published:** 2019-11-10

**Authors:** Daniel Hernandez-Patlan, Bruno Solis-Cruz, Karine P. Pontin, Juan D. Latorre, Xochitl Hernandez-Velasco, Ruben Merino-Guzman, Abraham Mendez-Albores, Billy M. Hargis, Raquel Lopez-Arellano, Guillermo Tellez-Isaias

**Affiliations:** 1Laboratorio 5, LEDEFAR, Unidad de Investigacion Multidisciplinaria, Facultad de Estudios Superiores (FES) Cuautitlan, Universidad Nacional Autonoma de Mexico (UNAM), Cuautitlan Izcalli 54714, Mexico; danielpatlan@comunidad.unam.mx (D.H.-P.); rlajjd@yahoo.com.mx (R.L.-A.); 2Departamento de Medicina Veterinária Preventiva, Centro de Diagnóstico e Pesquisa em Patologia Aviária, Universidade Federal do Rio Grande do Sul, Porto Alegre RS 97105-900, Brazil; pontin.karine@gmail.com; 3Department of Poultry Science, University of Arkansas, Fayetteville, AR 72704, USA; juandlatorre@gmail.com (J.D.L.); bhargis@uark.edu (B.M.H.); 4Departamento de Medicina y Zootecnia de Aves, Facultad de Medicina Veterinaria y Zootecnia, UNAM, Ciudad de Mexico 04510, Mexico; xochitl_h@yahoo.com (X.H.-V.); onirem@unam.mx (R.M.-G.); 5Laboratorio 14, Alimentos, Micotoxinas y Micotoxicosis, Unidad de Investigacion Multidisciplinaria, FES Cuautitlan, UNAM, Cuautitlan Izcalli 54714, Mexico; albores@unam.mx

**Keywords:** Ascorbic acid, curcumin, chickens, *Salmonella Enteritidis*, intestinal integrity

## Abstract

Two experimental models were conducted to evaluate and compare the effect of ascorbic acid (AA) or curcumin formulated in a solid dispersion (SD-CUR) as prophylactic or therapeutic alternatives to prevent or control *S.* Enteritidis (SE) infection in broiler chickens. In the prophylactic model, dietary administration of AA showed a significant reduction in SE counts in crop compared to the positive control (PC) group (*p* < 0.05), whereas in cecal tonsils (CT), SD-CUR significantly reduced SE recovery. Superoxide dismutase (SOD) activity was significantly higher in chickens supplemented with AA or SD-CUR, and total intestinal IgA levels were significantly lower in both treatments when compared to the PC group. Serum fluorescein isothiocyanate-dextran (FITC-d) levels were reduced by SD-CUR compared to PC, while AA presented significantly lower total aerobic bacteria. In the therapeutic model, only the dietary administration of AA significantly decreased SE in crop and CT on days 3 and 10 post-challenge. FITC-d levels were significantly lower in both treated groups in comparison to PC, but IgA levels were significantly reduced only by AA. The results suggest that dietary AA and SD-CUR have different modes of action to reduce SE intestinal colonization in two different challenge models in broiler chickens.

## 1. Introduction

The restriction of antibiotics at subtherapeutic doses in animal production as growth promoters has been associated with increased bacterial infections in poultry and, paradoxically, greater use of antibiotics [1,2,3,4,5,6]. Therefore, interest in finding viable alternatives with similar benefits to antibiotics has increased in recent years [7], mainly to prevent, control, and treat infections associated with *Salmonella*, a foodborne pathogen that remains a significant concern in public health [8]. In recent years, the investigation of alternatives to antibiotics has been focused on improving intestinal health through the use of feed additives such as probiotics, prebiotics, in-feed enzymes, essential oils, herbal extracts, and antioxidants [9].

In poultry production, antioxidants are included in diets primarily to protect feed from degradation and deterioration during storage, as well as for nutritional purposes [10]. However, it has been reported that these additives play an essential role in the prevention of several diseases in poultry due to their different mechanisms of action [11]. 

Ascorbic acid (AA) is a water-soluble organic compound with potent antioxidant properties due to its ability to readily donate electrons to protect the host from oxidative stress [12,13]. Furthermore, AA has an immunomodulatory effect and can improve the microbial diversity and function [13,14]. Dietary supplementation with AA has shown to promote positive effects in reducing the physiological stress caused by the rapid growth rate and the ever-changing environmental conditions in poultry production [15]. 

Meanwhile, Curcumin, a polyphenolic compound derived from turmeric, a product of the plant *Curcuma longa* [16], has been widely used in the poultry industry as an anticoccidial, anti-inflammatory, immunomodulatory, antimicrobial, antioxidant, and to promote growth performance [17,18,19]. Nevertheless, some limitations of curcumin are its poor aqueous solubility and intestinal permeability [20]. Due to these limitations, the preparation of solid dispersions of curcumin (SD-CUR) has managed to increase both the solubility and permeability of curcumin given the fact that the crystalline structure changes to an amorphous form [21]. Therefore, in the present study, two natural antioxidants such as AA and SD-CUR were evaluated as prophylactic or therapeutic alternatives to prevent or control *S.* Enteritidis infection and help to maintain healthy intestinal integrity in broiler chickens.

## 2. Results

Results of the dietary administration of AA and SD-CUR at 0.1% in the feed as prophylactic agents on *S*. Enteritidis counts are shown in Table 1. In both independent trials, *S*. Enteritidis counts in crop were statistically 1.2 log lower in the AA group compared to the PC group (*p* < 0.05), whereas in CT, chickens supplemented with SD-CUR presented significantly lower *S*. Enteritidis counts and *S*. Enteritidis incidence (*p* < 0.05) in comparison with the PC group. Furthermore, SOD activity was significantly higher in chickens supplemented with AA and SD-CUR in comparison with the PC group, whereas total intestinal IgA levels were significantly lower in the AA and SD-CUR groups compared to the PC group (Table 2). However, serum FITC-d levels were only significantly lower in the SD-CUR group when compared to the PC group (Table 2). Finally, TAB were significantly reduced in chickens treated with AA compared to the other two experimental groups, but chickens treated with SD-CUR tended to increase the counts of TAB (P = 0.07) compared to the animals consuming the control unsupplemented diet (PC).

Table 3 shows the bacterial counts and incidence of *S*. Enteritidis after 3 and 10 days of treatment with AA and SD-CUR at 0.1% inclusion in the feed post-*S.* Enteritidis challenge (therapeutic model). On day three post-*S.* Enteritidis challenge, *S.* Enteritidis counts in the crop, and CT significantly decreased by 2.05 log and 3.54 log in the AA in relation to control, resulting in a significant decrease in the incidence of *S*. Enteritidis in both crop (*p* < 0.01) and CT (*p* < 0.005). Interestingly, although not significant, *S*. Enteritidis counts in the crop and CT were reduced by 1.02 log and 1.59 log in the SD-CUR group compared to the PC group. Meanwhile, on day 10 post-*S*. Enteritidis challenge, although the *S*. Enteritidis counts in crop were significantly lower in the AA and SD-CUR groups when compared to the PC group, only the AA group significantly decreased the *S*. Enteritidis counts in CT by 4.71 log, as well as the incidence of *S*. Enteritidis (*p* < 0.001). Furthermore, serum FITC-d levels were significantly lower in AA and SD-CUR groups in comparison with the PC group (Table 4). However, only total intestinal IgA levels were statistically lower in the AA group compared to the PC group. Finally, no significant differences in SOD activity were found among the experimental groups (Table 4).

## 3. Discussion

The production of antibiotic-free poultry is a worldwide trend [22] derived from the restriction of the use of antibiotics as a measure to reduce the problems of bacterial resistance and maintain the safety of food [6]. However, this measurement has increased the incidence of bacterial infections [2]. Therefore, the interest in finding viable alternatives with similar benefits to antibiotics has increased in recent years [7], mainly to reduce bacterial resistance problems, improve or maintain performance parameters, and control foodborne pathogens such as *Salmonella* [8]. 

Dietary AA supplementation into the feed at 0.1% in the prophylactic model (Table 1) proved to be more effective in reducing the *S*. Enteritidis counts in the crop than in the CT, in both trials. This is due to the capability of AA, a weak acid (pKa = 4.1 and 11.6), to reduce the pH in the crop by the release of protons [23,24]. However, as AA begins to degrade as the pH increases [25], its acidifying capacity is affected, as well as its influence on the stimulation of growth of beneficial bacteria like *Lactobacillus* and *Bifidobacterium* [26]. Therefore, these results support our findings in CT and agree with other published studies in which supplementation with 1% AA in the feed had no effect reducing the intestinal pH of broilers [27,28]. In contrast, supplementation with SD-CUR significantly reduced *S*. Enteritidis counts in CT by > 1.5 log in both independent trials (Table 1). This reduction is related to the increase in the solubility and permeability of curcumin when formulated in a solid dispersion using PVP-K30 [29,30] and, therefore to the improvement of its antimicrobial and immunoregulatory-immunostimulatory effects [19,31,32]. We have observed in an in vitro study previously published by our laboratory that raw curcumin had no antimicrobial effect against *S.* Enteritidis [33]. Additionally, it has been shown that curcumin supplementation promotes changes in the composition and diversity of the gut microbiome since the relative abundance of the *Lactobacillus* genus could be increased [34,35,36]. This increase in the abundance of *Lactobacillus* has a beneficial effect in suppressing the growth of pathogenic bacteria in the intestine of chickens [37] due to their different antimicrobial mechanisms [38,39], as well as, the maintenance of intestinal homeostasis. Although lactic acid bacteria were not determined in the present study, as shown in Table 2, only chickens treated with SD-CUR increased the total anaerobic bacteria counts (0.31 log_10_ cfu/gr, P = 0.07), which suggests an increase in the diversity of the microbiota that could be related to a higher activity against *Salmonella*.

Previous studies have demonstrated that *Salmonella* can disrupt intercellular junctions, causing an increase of paracellular permeability and bacterial translocation to facilitate its pathogenicity [40]. In this sense, a way to evaluate the intestinal integrity is the determination of the serum FITC-d levels, a molecule that under normal intestinal conditions does not cross the mucosal barrier due to its large size (3–5 kDa) [41,42]. The results showed that the group supplemented with SD-CUR presented significantly lower serum FITC-d levels (0.159 μg/mL) compared to the other experimental groups. Lower serum FITC-d levels are related to the maintenance of intestinal integrity due to the reduction of *Salmonella* counts and the ability of curcumin to restore the intestinal barrier function and expression of tight junction proteins, as well as the proliferation/regeneration of the intestinal epithelium [35,43,44]. 

Mucosal immunity provides the first line of defense against oral exposure to pathogens, avoiding their adherence and invasion of epithelial cells [45]. *Salmonella* infection can induce a rapid and robust local inflammatory response in the intestinal epithelium, leading to the secretion of pro-inflammatory cytokines IL-1, IL-6, IL-23 IL-12, and IL-18, these last two lead to the production of interferon-gamma (IFN-γ) and tumor necrosis factor-alpha (TNF-α) followed by the production of IgA and antimicrobial peptides (defensins, cathelicidins, histatins, and lactoferrins) as a defensive mechanism to limit the mucosal colonization of pathogens [45,46,47,48,49]. In both treatment groups, total intestinal IgA levels were significantly lower compared to the PC group (AA: 2.7 μg/mL and SD-CUR: 3.01 μg/mL), which is consistent with the lower count of *S*. Enteritidis and their possible positive effects on the maintenance of intestinal integrity due also to their anti-inflammatory and antioxidant properties since it has been described that the production is not immediate [45,50]. It has been reported that curcumin does not only down-regulate pro-inflammatory cytokines to reduce local inflammation in the intestine, but also reduces systemic inflammation triggered by the release of lipopolysaccharide (LPS) into circulation [44,51,52]. Similar to curcumin, AA is capable of decreasing the expression of proinflammatory cytokines such as IL-1, IL-6, IL-12, TNF-α and IFN-γ [53,54], which is indicative of better intestinal health.

The results obtained of total intestinal IgA levels in the AA and SD-CUR groups are also supported by the significant increase in the antioxidant activity of SOD compared to the PC group (0.92 and 0.90 U/mL, respectively) since the increase in antioxidant capacity can lead to the reduction of oxidative stress and inflammation [9]. SOD is one of the most important antioxidant enzymes involved in the protection of tissues from oxidative damage by regulating various reactive oxygen and nitrogen species (ROS/RNS) [55]. However, the increase in SOD activity in the groups treated with AA and SD-CUR was due to their ability to stimulate the production of this antioxidant enzyme to protect the host against oxidative stress and lipid peroxidation [56,57,58].

Unlike the prophylactic model, in the therapeutic model, only the dietary administration of AA significantly decreased the counts and incidence of *S*. Enteritidis in both the crop and CT on days 3 and 10 post-*S*. Enteritidis challenge. These results are related to the acidifying capacity of AA in the crop [23,24]; whereby the concentration of *S*. Enteritidis that reached the intestinal epithelium was much lower. Furthermore, these results are supported by those previously published by our research group where AA was able to reduce the concentration of *S*. Enteritidis in the compartment that simulates the crop in an in vitro avian digestion model [33]. Although not significant either on days 3 and 10 post-*S.* Enteritidis challenge, *S*. Enteritidis counts in CT were reduced in chickens treated with SD-CUR by 1.59 log and 1.80 log, respectively. This could be because the dose of SD-CUR was insufficient to exert a potent antimicrobial activity in this therapeutic model, which indicates that SD-CUR has a better effect in preventing *S*. Enteritidis infections/colonization. Despite these results, serum FITC-d levels were reduced by AA and SD-CUR treatments. In the SD-CUR treated group, the decrease in the serum FITC-d levels is due to the ability of curcumin to restore the intestinal barrier function and expression of tight junction proteins, resulting in a reduction of paracellular permeability [35,52]. Meanwhile, in the treatment with AA, the reduction of mucosal FITC-d permeability is mainly associated with the decrease in the severity of *S*. Enteritidis infection [59], which was also reflected in the lower total intestinal IgA levels compared to the SD-CUR and PC groups. However, the SD-CUR treated group had lower IgA values compared to the PC group, but not significantly. Finally, there were no significant differences in SOD activity among groups. The high activity of SOD in the treated groups is related to the capability of these two antioxidants to stimulate the production of antioxidant enzymes. However, the slight increase in SOD in the PC group was because this antioxidant enzyme protects the tissues from the oxidative damage of ROS/RNS, a defense mechanism against microbial invasion and replication [55].

## 4. Materials and Methods 

### 4.1. Preparation of Experimental Treatments and Diets

Two treatments were evaluated: (1) AA (99%–100%, Food grade, Drogueria Cosmopolitan, Naucalpan, Edo. de Mex., Mexico); and (2) a solid dispersion of curcumin (SD-CUR). The first treatment was prepared by granulating 90% AA with 10% microcrystalline cellulose (MCC, Avicel® PH 102), followed by a drying step and subsequent sieving. Treatment 2 was prepared by dissolving 1 part of curcumin in 9 parts of a polyvinylpyrrolidone (PVP) K30 solution, followed by water evaporation at 40 ºC and sieving. In both treatments, the sieving was done using a No. 25 mesh to obtain particles of around 700 µm. These treatments were included in starter basal diets without antibiotics, coccidiostats, or enzymes at a concentration of 0.1% (1 kg/Ton of feed). A starter diet was formulated to approximate the nutritional requirements of broiler chickens as recommended by the National Research Council [60] and adjusted to breeder’s recommendations [61]. All animal handling procedures complied with the Institutional Animal Care and Use Committee (IACUC) at the University of Arkansas, Fayetteville (protocol #18029).

### 4.2. Salmonella Strain and Culture Conditions

A primary poultry isolate of *Salmonella enterica* serovar Enteritidis bacteriophage type 13A, was obtained from the USDA National Veterinary Services Laboratory (Ames, IA, USA). This strain is resistant to 25 µg/mL of novobiocin (NO, catalog no. N-1628, Sigma) and was selected due to its resistance to 20 µg/mL of nalidixic acid (NA, catalog no. N-4382, Sigma) in our laboratory. The *Salmonella* Enteritidis culture was performed according to previous publications [19] to obtain approximate bacterial concentrations of 4 × 10^4^ and 4 × 10^7^ cfu/mL. Levels of *S*. Enteritidis were further verified by serial dilutions and plated on brilliant green agar (BGA, Catalog No. 70134, Sigma) with NO and NA for enumeration of actual cfu used in the experiment.

### 4.3. Experimental Design 

Two experiments were conducted to evaluate the effect of AA and SD-CUR at 0.1% in the feed in a prophylactic or therapeutic model against *S*. Enteritidis infection and their influence on intestinal integrity in broiler chickens.

#### 4.3.1. Prophylactic Model

The prophylactic model consisted of two independent trials with 45 day-of-hatch Cobb-Vantress male broiler chickens (Fayetteville, AR, USA) each. In both trials, chickens were randomly assigned to one of three groups (n = 15 chickens/group): (1) positive control group (PC), (2) 0.1% AA in the feed, and (3) 0.1% SD-CUR in the feed. Chicks were housed in brooder battery cages, provided with their respective diet and water ad libitum, as well as maintained at an age-appropriate temperature during the experiment. On day 6 of age, all chicks were orally challenged with 1 × 107 cfu of S. Enteritidis per bird and weighed to calculate the concentration of fluorescein isothiocyanate-dextran (FITC-d) to be administered according to the group body weight (only trial 2). Subsequently, 24 h post-S. Enteritidis challenge (7 day-old), chicks were euthanized by CO2 inhalation, and samples of crop and cecal tonsils (CT) for S. Enteritidis colony counts from 12 broilers per group were collected (both independent trials). Additionally, only in trial 2, blood samples from the femoral vein for determination of FITC-d and superoxide dismutase (SOD) as described below, as well as samples of CT for total aerobic bacteria (TAB) colony counts and intestinal samples for total intestinal IgA levels (n = 12/group) were also collected.

#### 4.3.2. Therapeutic Model

To evaluate this model, 90 one-day-old Cobb-Vantress male broiler chickens (Fayetteville, AR, USA) were challenged with 1 × 10^4^ cfu of *S*. Enteritidis per bird at day of hatch and randomly allocated to one of three groups (n = 30 chickens): (1) positive control group (PC); (2) 0.1% AA in the feed; and (3) 0.1% SD-CUR in the feed. Chicks were housed in brooder battery cages, provided with their respective diet and water ad libitum, as well as maintained at an age-appropriate temperature during the experiment. On days 3 and 10 post-*S*. Enteritidis challenge, 15 chicks from each group were euthanized by CO_2_ inhalation, respectively, but only the crop and CT from 12 birds per group were aseptically collected for *S*. Enteritidis count. Blood samples were also collected from the femoral vein for the determination of FITC-d and SOD, only on day 10 post-*S*. Enteritidis challenge. The concentration of FITC-d administered was calculated based on group body weight at day 9 post-*S*. Enteritidis challenge. Furthermore, intestinal samples for total intestinal IgA levels were also collected at day 10 post-challenge.

### 4.4. Salmonella and Total Aerobic Bacteria (TAB) Counts 

In both experimental models, the crop and CT samples from 12 chickens per group were homogenized and diluted with saline (1:4 *w*/*v*), and 10-fold dilutions were plated on BGA with NO and NA for *S*. Enteritidis counts or on Tryptic Soy Agar (TSA, catalog no. 211822, Becton Dickinson, Sparks, MD) for TAB determination (only in the prophylactic model, trial 2). Plates were incubated at 37 °C for 24 h to enumerate total *S*. Enteritidis and TAB colony-forming units. Subsequently, the crop and CT samples were enriched in 2× concentrated tetrathionate enrichment broth and further incubated at 37 °C for 24 h. Enrichment samples were streaked onto Xylose Lysine Tergitol-4 (XLT-4, Catalog No. 223410, BD Difco^TM^) selective media for confirmation of *Salmonella* incidence.

### 4.5. Serum FITC-d Levels

FITC-d (MW 3–5 kDa; Sigma-Aldrich Co., St. Louis, MO, USA) was provided by oral gavage to 12 broiler chickens from each group at a dose of 8.32 mg/kg of body weight one hour before the chicks were euthanized by CO_2_ inhalation in order to collect blood samples and evaluate the paracellular transport and mucosal barrier dysfunction [62,63]. Three remaining broiler chickens of each group were used as controls. The blood samples were centrifuged (1000× *g* for 15 min) to separate the serum. Then, serum samples obtained were diluted (1:5) and measured fluorometrically at an excitation wavelength of 485 nm and an emission wavelength of 528 nm (Synergy HT, Multi-mode microplate reader, BioTek Instruments, Inc., VT, USA) to determine the serum FITC-d levels [41].

### 4.6. Superoxide Dismutase Activity

Superoxide dismutase (SOD) activity was measured in serum samples from 12 chickens using a commercial assay kit in each experimental model (Cayman chemical company, Item No. 706002, Ann Arbor, MI, USA) following the manufacturer’s instructions. Three types of SOD (Cu/Zn, Mn, and FeSOD) were determined, and the optimal dilution to quantify the SOD activity was 1:5. Samples were measured at 450 nm using an ELISA plate reader (Synergy HT, multi-mode microplate reader, BioTek Instruments, Inc., Winooski, VT, USA).

### 4.7. Total Intestinal Immunoglobulin A (IgA) Levels

In each experimental model, intestinal sections of 5 cm from the Meckel’s diverticulum to the ileocecal junction from 12 chickens per group were taken to quantify total IgA levels [64]. Briefly, intestinal sections were rinsed three times with 5 mL of 0.9% saline; then the rinse was collected in a tube and centrifuged at 1864× *g* at 4 °C for 10 min. Subsequently, the supernatants were separated and stored at −20 °C until tested. A commercial indirect ELISA kit was used to quantify IgA according to the manufacturer’s instructions (Catalog No. E30-103, Bethyl Laboratories Inc., Montgomery, TX 77356, USA). The intestinal rinse supernatants were diluted (1:100), placed in 96-well plates (Catalog No. 439454, Nunc MaxiSorp, Thermo Fisher Scientific, Rochester, NY, USA), and measured at 450 nm using an ELISA plate reader (Synergy HT, multi-mode microplate reader, BioTek Instruments, Inc., Winooski, VT, USA).

### 4.8. Data and Statistical Analysis

Data from *S*. Enteritidis and TAB counts (log cfu/g), serum FITC-d levels, total intestinal IgA levels, and SOD activity were subjected to analysis of variance (ANOVA) as a completely randomized design using the General Linear Models procedure of Statistical Analysis System (SAS®) [65]. Significant differences among the means were determined by Duncan’s multiple range test at P < 0.05. Enrichment data were expressed as positive/total chickens (%), and the percentage of *S*. Enteritidis positive samples were compared by a chi-squared test of independence [66], testing all possible combinations to determine the significance (P < 0.05).

## 5. Conclusions

The inclusion of AA and SD-CUR in the diet can be an alternative for the production of antibiotic-free poultry to reduce bacterial antimicrobial resistance problems and maintain food safety that is extremely important in public health concern. The results suggest that dietary AA or SD-CUR have different modes of action to reduce SE intestinal colonization in two different challenge models in broiler chickens. Further studies to confirm these results and using higher concentrations of these additives are currently being evaluated.

## Figures and Tables

**Table 1 pathogens-08-00229-t001:** *Salmonella* Enteritidis (SE) counts ^1^ and incidence ^3^ in crop and cecal tonsils (CT) in broiler chickens supplemented with ascorbic acid (AA) and a solid dispersion of curcumin (SD-CUR). Prophylactic model ^2^.

Treatments	Crop Log_10_ cfu/g	Crop + / - (%) ^3^	CT Log_10_ cfu/g	CT + / - (%)
		**Trial 1**		
**Ctrl**	2.68 ± 0.47 ^ab^	9/12 (75%)	4.01 ± 0.29 ^a^	12/12 (100%)
**AA**	1.48 ± 0.53 ^b^	5/12 (42%)	3.69 ± 0.17 ^a^	12/12 (100%)
**SD-CU**	3.08 ± 0.57 ^a^	9/12 (75%)	2.42 ± 0.54 ^b^	8/12 (67%) *
		**Trial 2**		
**Ctrl**	2.69 ± 0.48 ^a^	9/12 (75%)	3.94 ± 0.22 ^a^	12/12 (100%)
**AA**	1.49 ± 0.54 ^b^	5/12 (42%)	3.80 ± 0.28 ^ab^	12/12 (100%)
**SD-CUR**	3.19 ± 0.47 ^a^	9/12 (75%)	2.34 ± 0.50 ^b^	8/12 (67%) *

^1^ Data expressed in Log_10_ cfu /g of tissue. Mean ± standard error from 12 chickens (*p* < 0.05). ^2^ Chickens were orally gavaged with 10^7^ cfu of *S.* Enteritidis per chicken at 6-d old, samples were collected 24 h later. ^3^ Data expressed as positive/total chickens (%). * *p* < 0.05. ^a–b^ Values within treatment columns for each treatment with different superscripts differ significantly (*p* < 0.05).

**Table 2 pathogens-08-00229-t002:** Determination of total aerobic bacteria (TAB), serum fluorescein isothiocyanate–dextran (FITC-d) levels, superoxide dismutase (SOD) activity and total intestinal IgA levels in broiler chickens treated with ascorbic acid (AA) and a solid dispersion of curcumin (SD-CUR) in the prophylactic model (Trial 2) ^1^.

Treatments	TAB Log_10_ cfu/g	FITC-d (μg/mL)	SOD (U/mL)	IgA (μg/mL)
**Ctrl**	7.96 ± 0.10 ^ab^	0.591 ± 0.055 ^a^	3.58 ± 0.31 ^b^	14.21 ± 0.83 ^a^
**AA**	7.92 ± 0.11 ^b^	0.533 ± 0.034 ^ab^	4.50 ± 0.35 ^a^	11.51 ± 0.71 ^b^
**SD-CUR**	8.27 ± 0.13 ^a^	0.432 ± 0.037 ^b^	4.48 ± 0.20 ^a^	11.20 ± 0.53 ^b^

^1^ Data are presented as mean ± standard error from 12 chickens (*p* < 0.05). ^a–b^ Values within treatment columns for each treatment with different superscripts differ significantly (*p* < 0.05).

**Table 3 pathogens-08-00229-t003:** *Salmonella* Enteritidis (SE) counts ^1^ and incidence ^3^ in crop and cecal-tonsils (CT) in broiler chickens supplemented with ascorbic acid (AA) and a solid dispersion of curcumin (SD-CUR) in the therapeutic model ^2^.

Treatments	Crop Log_10_ cfu/g	Crop + / − (%)	CT Log_10_ cfu/g	CT + / − (%)
	**3-d post—*S.* Enteritidis Challenge**			
**Ctrl**	3.18 ± 0.46 ^a^	10/12 (83%)	6.44 ± 0.15 ^a^	12/12 (100%)
**AA**	1.13 ± 0.48 ^b^	4/12 (33%) *	2.90 ± 0.91 ^b^	6/12 (50%) **
**SD-CUR**	2.16 ± 0.46 ^ab^	8/12 (67%)	4.85 ± 0.86 ^ab^	9/12 (75%)
	**10-d post—*S.* Enteritidis Challenge**			
**Ctrl**	2.93 ± 0.65 ^a^	7/12 (58%)	6.61 ± 0.21 ^a^	12/12 (100%)
**AA**	1.26± 0.54 ^b^	4/12 (33%)	1.89 ± 0.81 ^b^	4/12 (33%) φ
**SD-CUR**	0.97 ± 0.51 ^b^	3/12 (25%)	4.81 ± 0.85 ^ab^	9/12 (75%)

^1^ Data expressed in Log_10_ cfu /g of tissue. Mean ± standard error from 12 chickens (*p* < 0.05). ^2^ Chickens were orally gavaged with 10^7^ cfu of *S.* Enteritidis per chicken at 6-d old, samples were collected 24 h later. ^3^ Data expressed as positive/total chickens (%). * *p* < 0.01; ** *p* < 0.005; φ *p* < 0.001. ^a–b^ Values within treatment columns for each treatment with different superscripts differ significantly (*p* < 0.05).

**Table 4 pathogens-08-00229-t004:** Determination of serum fluorescein isothiocyanate–dextran (FITC-d) levels, superoxide dismutase (SOD) activity and total intestinal IgA levels in broiler chickens supplemented with ascorbic acid (AA) and a solid dispersion of curcumin (SD-CUR). Therapeutic model (day 10 post-*S*. Enteritidis challenge) ^1^.

Treatments	FITC-d (μg/mL)	SOD (U/mL)	IgA (μg/mL)
**Ctrl**	0.700 ± 0.020 ^a^	10.34 ± 0.67 ^a^	14.34 ± 2.81 ^a^
**AA**	0.457 ± 0.039 ^b^	10.22 ± 0.72 ^a^	9.18 ± 2.95 ^b^
**SD-CUR**	0.489 ± 0.020 ^b^	9.72 ± 0.82 ^a^	11.26 ± 3.39 ^ab^

^1^ Data are presented as mean ± standard error from 12 chickens (*p* < 0.05). ^a–b^ Values within treatment columns for each treatment with different superscripts differ significantly (*p* < 0.05).

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
