# Peer review of "Evaluation of Ascorbic Acid or Curcumin Formulated in a Solid Dispersion on Salmonella Enteritidis Infection and Intestinal Integrity in Broiler Chickens"

_pathogens, 2019, doi:10.3390/pathogens8040229_

Round 1
Reviewer 1 Report
Major comments:
It is a well-designed study. Please revise the errors and fix the logical problems below.
Abstract: please clarify the first model in the abstract. No challenge or raising in a commercial condition? It is confusing that results are different in different models.
Please revise your conclusion. What is the difference between “against Salmonella” and “Salmonella Infection?” If you want to prove that AA could be used as a treatment, you will need another control that birds receive no challenge. Then the AA should help birds to recover the lesion or any other immune levels to those observed in the negative control. Please specify this statement and let your data support this statement.
Introduction:
Line 43 to 46: the sentence needs English editing. Since the restriction…, not only infection has been increasing but also …purposes, which could generate to … It is not a complete sentence.
Results:
It is confusing how many trials you have run. In Table 2, you demonstrated that the table generated from the prophylactic model trial 2. Did you repeat that prophylactic model?
Table 3. Please remove “trial”. It is the same trial but you collected birds at different ages, correct? It is confusing.
Discussion:
Line 133-134. Therapeutic doses for growth performance purposes in antibiotic-free poultry? The reviewer couldn’t follow the authors’ logic here. Firstly, only a sub-therapeutic dosage of antibiotics is used as growth promoters in the feed. Secondly, antibiotics are forbidden to antibiotic-free poultry.
Please remove the whole passage from line 131 to 140. The purpose or hypothesis of the study should be in the introduction instead of the discussion.
Line 147: AA in the feed had no effect reducing the intestinal pH? Authors contracted to themselves that in line 143: AA,… to reduce the pH in the crop. Please explain why AA works differently in the crop and intestinal tract.
Line 156 to 159. It is not a full sentence.
Line 181: When authors claimed that their products improved intestinal integrity. I was expecting to see the morphology results instead of the immune level changes. Please be scientific and accurate with the writing.
Materials and Methods:
How did the authors distinguish the prophylactic model and therapeutic model? In your study, the therapeutic model used a lower challenge SE dosage. Is the challenge SE pathogenic to broiler chicken? Any virulent test? As I knew, most wild SE strains are not pathogenic to chicken. Chickens are more like carriers of SE. Please list references or any standards about the challenge model.
Author Response
"Please see the attachment"

Reviewer 2 Report
Dear Authors,
After the review process, I have several comments:
you should insert references in all Materials and Methods sections; you should insert references for all activities mentioned in Page 2, Line 64-65; you should insert new references about curcumin published in mdpi journals, for example: https://doi.org/10.3390/pharmaceutics11040191; you should insert comments about the level of curcumin and intestinal integrity (stability is an important aspect); you should comment the limitation of the study in Discussion section, based on the antimicrobial capacity of the solid dispersion products. It is not clear what determined the antimicrobial effects, based on products (substrated) or other compounds resulted after the biotransformations process (possible); you should change ″... ascorbic acid (AA) or curcumin formulated ...″with ″ascorbic acid (AA) and curcumin formulated″, because it generates confusion in the paper.
Best regards!
Author Response
"Please see the attachment"

Round 2
Reviewer 2 Report
Dear Authors,
After the second review, I do not have any other comments.
Best regards!